# Prognostic Value of Salivary Biochemical Indicators in Primary Resectable Breast Cancer

**DOI:** 10.3390/metabo12060552

**Published:** 2022-06-16

**Authors:** Lyudmila V. Bel’skaya, Elena A. Sarf

**Affiliations:** Biochemistry Research Laboratory, Omsk State Pedagogical University, 14 Tukhachevsky str, 644043 Omsk, Russia; nemcha@mail.ru

**Keywords:** saliva, biochemistry, primary resectable breast cancer, prognosis, overall survival, recurrence

## Abstract

Despite the fact that breast cancer was detected in the early stages, the prognosis was not always favorable. In this paper, we examined the impact of clinical and pathological characteristics of patients and the composition of saliva before treatment on overall survival and the risk of recurrence of primary resectable breast cancer. The study included 355 patients of the Omsk Clinical Oncology Center with a diagnosis of primary resectable breast cancer (T_1-3_N_0-1_M_0_). Saliva was analyzed for 42 biochemical indicators before the start of treatment. We have identified two biochemical indicators of saliva that can act as prognostic markers: alkaline phosphatase (ALP) and diene conjugates (DC). Favorable prognostic factors were ALP activity above 71.7 U/L and DC level above 3.93 c.u. Additional accounting for aspartate aminotransferase (AST) activity allows for forming a group with a favorable prognosis, for which the relative risk is reduced by more than 11 times (HR = 11.49, 95% CI 1.43–88.99, *p* = 0.01591). Salivary AST activity has no independent prognostic value. Multivariate analysis showed that tumor size, lymph nodes metastasis status, malignancy grade, tumor HER2 status, and salivary ALP activity were independent predictors. It was shown that the risk of recurrence decreased with menopause and increased with an increase in the size of the primary tumor and lymph node involvement. Significant risk factors for recurrence were salivary ALP activity below 71.7 U/L and DC levels below 3.93 c.u. before treatment. Thus, the assessment of biochemical indicators of saliva before treatment can provide prognostic information comparable in importance to the clinicopathological characteristics of the tumor and can be used to identify a risk group for recurrence in primary resectable breast cancer.

## 1. Introduction

Breast cancer is the second most common cancer after lung cancer [1,2]. The high incidence of breast cancer is due to both genetic and environmental factors [3]. Despite the fact that the methods of treatment of breast cancer are constantly being improved and the mortality rate is decreasing, it is impossible to completely avoid regional and distant recurrences [3,4]. Early detection of primary and recurrent breast cancer is of great clinical importance for making breast cancer treatment decisions to improve survival rates [5]. Early (primarily operable) breast cancer includes stages 0, I, IIA, IIB, IIIA (T_1-3_N_0-1_M_0_) [6]. However, even if breast cancer is detected in the early stages, the prognosis is not always favorable.

Traditional prognostic factors for breast cancer are tumor size, axillary lymph node status, lymphatic and vascular invasion, hormone receptor status, and expression of human epidermal growth factor receptor 2 (HER2), but they do not fully reflect the prognosis of breast cancer [3,7,8]. Various serum tumor markers have been identified as prognostic, predictive factors in breast cancer patients, such as CEA, CA15-3, CA19-9, and CA125 [9,10]. Elevated levels of CA 15-3 or CEA have been shown to be associated with poor disease-free and overall survival [3,10,11]. The prognostic role of angiotensin-converting enzyme 2 (ACE2) has been described, elevated levels of which correlated with a decrease in disease-free survival [12]. Prognostic indicators based on inflammation are mentioned in the literature, including the ratio of neutrophils to lymphocytes, the ratio of lymphocytes to monocytes, the ratio of platelets to lymphocytes, and the ratio of C-reactive protein/albumin [10,13,14]. High levels of Ki-67 have also been significantly associated with poor survival [15]. Several studies have described the use of molecular markers characterizing the cellular pathways involved in tumor growth and spread (p53, RB, PI3K/Akt/mTOR, and Ras/MAPK) [16,17], as well as the tumor immune microenvironment, including tumor lymphoid infiltration (TILs) to determine the prognosis of breast cancer [18,19,20]. However, at present, none of the listed prognostic factors in blood serum are generally accepted in clinical practice.

A large number of studies are devoted to the use of saliva for the diagnosis of breast cancer [21,22,23,24,25,26,27,28,29,30,31]; however, the prognostic value of biochemical indicators of saliva has not yet been described. Previously, we have shown the potential applicability of biochemical indicators of saliva to determine the prognosis of lung cancer [32,33,34]. In particular, unfavorable signs are the level of imidazole compounds in the saliva of more than 0.478 mmol/L and lactate dehydrogenase activity of less than 545 U/L (HR = 4.17; 95% CI 1.36–12.51; *p* = 0.00000) [32]. We found that for different histological subtypes of lung cancer, different biochemical indicators of saliva are prognostically significant [33]. Due to the complete lack of literature data on the prognostic significance of biochemical indicators of saliva in breast cancer, we conducted a new selection of prognostically important signs. In this work, we tested the effect of 34 biochemical indicators of saliva on the overall survival of patients with primary resectable breast cancer. We also examined the impact of clinicopathological characteristics of patients and the composition of saliva before treatment on the risk of recurrence of primary resectable breast cancer.

## 2. Results

### 2.1. Overall Survival Rates Depending on the Clinicopathological Characteristics of Patients and Type of Treatment

During the follow-up, 63 (17.7%) patients died; 59 (16.6%) patients had a disease relapse. Overall survival and relative risk are shown in Table 1. A statistically significant factor of poor prognosis was the age of patients older than 70, disease stages pT_2_ and pT_3_, lymph node involvement pN_1_, high grade of G3 malignancy, luminal B-like (HER2−) molecular biological tumor subtype (Figure 1). Favorable prognostic factors include the HER2-positive status of breast cancer (Table 1). It was found that the presence of menopause was not a predictive factor of the disease nor the histological type of breast cancer (Table 1). There was no statistically confirmed improvement in overall survival with a positive ER and PR status.

It was found that the amount of surgical treatment did not significantly affect overall survival (Table 2); however, the relative risk was significantly higher in groups of patients who underwent radiation and/or chemotherapy. A risk reduction was observed for patients who received hormone therapy after treatment (Table 2).

### 2.2. Overall Survival Rates Depending on the Biochemical Composition of Saliva

We have identified three biochemical indicators of saliva that can act as prognostic markers, namely: alkaline phosphatase (ALP), aspartate aminotransferase (AST), and diene conjugates (DC) (Appendix A). As a threshold value, the medians of the corresponding indicators were used: for ALP—71.7 U/L, for DC—3.93 c.u., for AST—6.33 U/L (Appendix A). The results of the calculation of overall survival in groups, taking into account the threshold values of indicators, are shown in Table 3.

It was found that a favorable prognosis of breast cancer was associated with ALP activity above 71.7 U/L (Figure 2A), while the values of three-year survival were 90.4 and 95.3%, five-year survival—76.3 and 91.1%, seven-year survival 59.0 and 83.0% with an unfavorable and favorable prognosis, respectively. The content of diene conjugates is more than 3.93 c.u. was also a prognostically favorable sign (Figure 2C); the corresponding three-, five- and seven-year survival rates were 93.6, 87.0, and 75.7% in case of a favorable prognosis, respectively, and 91.3, 79.6 and 67.8% in case of unfavorable prognosis. AST activity had no independent prognostic value (Figure 2B); however, in combination with ALP activity, it made it possible to refine the prognosis of breast cancer (Figure 2E). The combination of ALP activity above 71.7 U/L and AST above 6.33 U/L was a prognostically favorable sign, while values below the threshold were unfavorable. All other combinations were intermediate. Differences in three-, five- and seven-year survival between groups with favorable and unfavorable prognosis, in this case, were more significant; the corresponding values were 96.8, 91.1, and 91.1% for a favorable prognosis and 89.4, 72.9, and 48.2% for poor prognosis, respectively. The simultaneous recording of ALP and AST activity made it possible to single out a group with the least favorable prognosis (overall survival—48.2%). Multivariate analysis of all three biochemical salivary indicators showed that AST activity was not an independent prognostic factor (Appendix A). Similarly, groups with a favorable prognosis were obtained with a combination of ALP activity above 71.7 U/L and the level of diene conjugates above 3.93 c.u. and an unfavorable prognosis with a combination of ALP activity below 71.7 U/L and the level of diene conjugates below 3.93 c.u. (Figure 2E). In this case, three-, five-, and seven-year survival rates were 96.4, 94.6, and 92.4% with a favorable prognosis, and 88.8, 72.3, and 61.6%, respectively, with an unfavorable prognosis. With a combination of all three favorable factors, the relative risk decreased by more than 11 times (Table 3). In this case, the survival rate was 96.6%, with a favorable prognosis throughout the entire follow-up period, with an unfavorable prognosis three-, five-, and seven-year survival rates were 90.5, 69.5, and 56.6%, respectively.

At the next stage, we conducted a multivariate analysis, which included parameters whose contribution to the one-way analysis was statistically significant (Table 4). Tumor size, lymph nodes metastasis status, malignancy grade, tumor HER2 status, and salivary ALP activity were shown to be independent prognostic features.

### 2.3. Analysis of the Risk of Relapse in Patients with Primary Operable Breast Cancer

An unfavorable prognosis for primary resectable breast cancer is associated primarily with the recurrence of the disease. Thus, 59 patients had a relapse during the observation period. We analyzed the main factors that could be associated with an increased risk of disease recurrence (Table 5). The risk of recurrence has been shown to decrease with menopause and increase with primary tumor size and lymph node involvement. The absence of radiation and chemotherapy reduced the risk of recurrence. A significant risk factor for recurrence was the activity of salivary alkaline phosphatase below 71.7 U/L before treatment and the level of DC below 3.93 c.u. We determined that the ALP activity in the saliva of patients with breast cancer recurrence was 60.8 [47.8; 78.2] U/L, while in the group without recurrence, alkaline phosphatase activity was higher—76.1 [48.9; 108.7] U/L. Differences between groups were statistically significant (*p* = 0.0274). The DC level in the groups with and without relapse was 3.89 [3.70; 4.10] and 3.93 [3.72; 4.17] c.u., respectively (*p* = 0.2529), so we used this indicator as an auxiliary one when calculating the prognosis in combination with ALP activity.

## 3. Discussion

We have shown that the factors of unfavorable prognosis are, first, the size of the tumor, the defeat of the lymph nodes and the high degree of malignancy, which is consistent with the literature data. It is known that despite a generally favorable prognosis for primary operable breast cancer, in cases of high malignancy or HER2 overexpression, survival rates decrease to 70–75% [35]. The invasion of the lymphatic vessels around the tumor significantly correlates with the size of the primary lesion, the histological malignancy of breast cancer, the involvement of the axillary lymph nodes, and the expression of hormone receptors [36]. The type of treatment was determined by the tumor’s corresponding clinical and pathological characteristics; therefore, it naturally affected the prognosis of breast cancer. Thus, patients with a large tumor size and lymph node involvement are subject to chemotherapy and radiation therapy and have worse overall survival rates (Table 2).

It has been shown for the first time that a number of biochemical indicators of saliva can be of prognostic value in breast cancer (Table 3). Thus, an independent prognostic factor is the activity of salivary alkaline phosphatase before treatment (Table 4). It is known that ALP is formed in the liver, skeletal tissue, intestines, kidneys, placenta, and various tumors and is usually considered a serum marker of hepatobiliary pathology and fractures [37]. Previous studies have shown that ALP is associated with systemic inflammation and tumor development [38,39]. ALP is a hydrolytic enzyme that dephosphorylates various types of molecules, including nucleotides, proteins, and alkaloids [40]. It plays an anti-inflammatory and tissue-protective role by enhancing the conversion of ATP to adenosine and increasing the level of adenosine [41]. A previous study showed that ALP activity is associated with cancer cell death, migration, and transition from mesenchyme to epithelium [39]. The albumin-to-alkaline phosphatase ratio (AAPR) is a combined measure associated with systemic inflammation, which is calculated by dividing the albumin level by the ALP level [42]. Higher serum ALP levels have been shown to be associated with a worse prognosis in nasopharyngeal carcinoma, prostate cancer, and colorectal cancer [43,44,45]. Chen et al. reported that pretreatment serum ALP was an independent adverse predictor of disease-free and overall survival in patients with basal-like breast cancer [46]. Women with breast cancer have ALP activities generally higher than in normal, healthy women. The progressive increase in the serum ALP activities with breast cancer indicates metastasis [47]. However, according to our data, increased ALP activity in saliva is, on the contrary, a favorable prognostic sign. We have previously shown that the activities of a number of enzymes in blood serum and saliva do not always coincide; therefore, the corresponding indicators in saliva should be considered independently and have their own reference values [48].

Salivary AST activity and the level of diene conjugates were additional prognostic indicators of saliva. According to current concepts, the effect of AST on cancer recurrence and survival is unclear [49]. Researchers’ understanding of the potential role of AST in human carcinogenesis remains speculative. Unlike normal cells, most cancer cells rely on aerobic glycolysis to generate the energy needed for cellular processes. Warburg suggested that mitochondrial dysfunction exists in cancer cells, as he observed that cancer cells could convert most glucose into lactate, regardless of the presence of oxygen [50]. Some studies have shown that cancer cell proliferation can also be powered by the metabolism of glutamine, which is required by tumor cells to support the biosynthesis of nucleotides and non-essential amino acids catalyzed by AST and ALT [51]. Thornburg et al. reported that oxamate could inhibit the proliferation of transformed mammary adenocarcinoma cells in vitro, and AST acts as an important metabolic target [52]. A number of studies have shown the predictive value of the De Ritis coefficient (AST/ALT-ratio) [53]. However, according to our data, it makes sense to consider only AST activity.

The pathogenetic role of oxygen free radicals and the processes of lipid peroxidation initiated by them in the development of diseases, including cancer, is widely known [34,54,55]. Oxidative stress manifests itself in the accumulation of damaged DNA bases, products of protein oxidation and lipid peroxidation, as well as in a decrease in the level of antioxidants and, as a result, an increase in the susceptibility of membrane lipids and lipoproteins to the action of prooxidants [56]. As the results of the observations showed, in the group of patients with primary operable breast cancer (stages I–IIA), the content of lipid peroxidation products was increased in the zone of breast neoplasia compared to the neighboring tissues of the gland—in the zone of the visible absence of malignant cells [57]. However, the prognostic role of DC has not yet been discussed. Prognostically favorable, according to our data, was an increase in the level of DC in saliva, which is due to a shift in the equilibrium of lipid peroxidation processes towards primary products (DC); against this background, the content of toxic Schiff bases and MDA decreases. Simultaneous assessment of ALP activity, AST, and DC level made it possible to form a group of patients with a favorable prognosis (all indicators were above the threshold values), for which the relative risk was 11.5 times lower compared to the group in which all of the listed indicators were below the threshold values (Table 3).

The picture of breast cancer recurrence has not changed significantly in recent years; however, there is a statistically significant increase in relapse-free survival. There is an improvement in the results of treatment of all types of breast cancer, especially HER2-positive and triple-negative, with a significant decrease in the frequency of early recurrences [58]. The likelihood of local recurrence is higher in patients with high-risk molecular subtypes [59,60]. However, a number of studies have shown that local recurrences after organ-preserving operations are not associated with a specific molecular subtype of the tumor [61,62]. In general, the incidence of local recurrence of breast cancer is from 10 to 30% and remains high after both breast-conserving operations and after mastectomy, despite the use of adjuvant radiation therapy [63]. We analyzed a group of patients with a local or distant breast cancer recurrence during the observation period (Table 5). It was shown that in the group with recurrence, there are more patients with preserved menstrual function (67.8% vs. 28.8%), a higher proportion of larger tumors (38.7% vs. 9.6% for pT2), and more often affected lymph nodes (67.3% vs. 26.0%). The remaining factors are not statistically significant factors in the occurrence of relapse. Additionally, it was shown that in the group with relapse of the disease, the activity of salivary alkaline phosphatase is lower than in the group without relapse. Moreover, the combination of ALP activity above 71.7 U/l and DC level above 3.93 c.u. is a favorable factor and reduces the risk of breast cancer recurrence by 2.5 times (Table 5).

The study’s limitations include the lack of information on anti-HER2 therapy performed in the study groups, as well as on Ki-67, which could provide additional information.

Thus, the assessment of biochemical indicators of saliva before treatment can provide prognostic information comparable in importance to the clinical and pathological characteristics of the tumor and can be used to identify a risk group for recurrence in primary resectable breast cancer.

## 4. Materials and Methods

### 4.1. Study Design and Group Description

The study included 355 patients of the Clinical Oncological Dispensary in Omsk, hospitalized with a diagnosis of primary resectable breast cancer in the period 2014–2017. Patients were enrolled after informed consent, and the study was performed following the approval from the ethical committee of the Omsk Regional Clinical Oncological Dispensary (21 July 2016, Protocol No. 15) and in accordance with Helsinki principles. All patients were divided into age groups with a step of ten years: 30–39 years old—34 (9.6%), 40–49 years old—68 (19.2%), 50–59 years old—117 (33.0%), 60–69 years old—105 (29.6%), over 70 years old—31 people (8.6%); 248 (69.9%) patients were in the postmenopausal state. For breast cancer staging, the AJCC TNM classification (8th edition, 2017) was used. Thus, pT_1_ stage was verified in 133 (37.5%) patients, pT_2_—in 172 (48.5%), pT_3_—in 50 (14.0%) patients. The status of lymph node involvement pN_0_ was confirmed in 245 (69.0%) patients and pN_1_ in the remaining 110 patients. According to the histological type, 171 (74.7%) patients had ductal breast cancer, 58 (25.3%) had lobular cancer. According to the histological degree of malignancy, G1 status was established for 28 (16.1%) patients, G2—for 58 (33.3%), G3—for 88 (50.6%) patients. According to the molecular biological subtype, 50 (15.9%) patients were assigned to luminal A-like, 41 (13.1%)—to luminal B-like (HER2-negative), 181 (57.6%)—to luminal B-like (HER2-positive), 22 (7.0%)—to Non-Luminal and 20 (6.4%)—to basal-like subtypes, respectively. According to the receptor status, HER2-negative status was 112 (35.0%), HER2-positive status—208 (65.0%), ER-negative—49 (15.3%), ER-positive—271 (84.7%), PR-negative—85 (26.6%), PR-positive—235 (73.4%) patients, respectively. Organ-preserving surgical treatment (sectoral resection) was performed in 61 (17.2%) patients, mastectomy in 286 (80.6%) patients, and in 8 (2.2%) patients; surgical treatment was not performed due to contraindications. Adjuvant radiation therapy was performed in 191 (53.8%) patients; adjuvant chemotherapy was performed in 181 (51.0%) patients and hormone therapy in 241 (67.9%) patients.

### 4.2. Determination of the Expression of the Receptors for Estrogen, Progesterone, and HER2

The Allred Scoring Guideline was used to assess the level of expression of estrogen receptors (ER), progesterone receptors (PR), and HER2 [64]. The level of expression of estrogen, progesterone, and HER2 receptors was assigned to one of four categories (−, +, ++, +++) in accordance with the ASCO/CAP recommendations [65]. According to the obtained values, breast cancer was classified into five groups: basal-like, luminal A-like, luminal B-like (HER2-negative), luminal B-like (HER2-positive), and Non-Luminal.

### 4.3. Collection and Analysis of Saliva

Saliva samples were collected at baseline, right before the start of treatment. Collection of saliva samples was carried out on an empty stomach after rinsing the mouth with water in the interval of 8–10 am by spitting into sterile polypropylene tubes. Saliva samples were centrifuged (10,000× *g* for 10 min) (CLb-16, Moscow, Russia), and biochemical analysis was performed without storage and freezing on a Stat Fax 3300 semi-automatic biochemical analyzer (Awareness Technology, Palm City, FL, USA) for 34 biochemical indicators, as described previously [28,66]. The full results of biochemical analysis of saliva for 34 indicators for a group of patients with primary resectable breast cancer compared with healthy controls are given in Appendix A. Appendix A shows the results of the biochemical analysis of saliva, depending on the presence/absence of relapse.

### 4.4. Statistical Analysis

The total follow-up time was 7 years; the median follow-up time was 59.8 months. The patient’s overall survival (OS) was assessed from the date of hospitalization to the date of the last observation (censored) or the date of death of the patient (complete). OS was assessed using the Kaplan–Meier method (Statistica 13.0, StatSoft, Tulsa, OK, USA). A univariate Cox proportional hazards regression analysis was initially carried out to investigate the relationships between salivary parameters and survival data (Appendix A). Finally, variables with *p* < 0.10 were chosen to formulate multivariate Cox proportional hazards regression models and determine the independent prognostic factors for OS (Appendix A).

## Figures and Tables

**Figure 1 metabolites-12-00552-f001:**
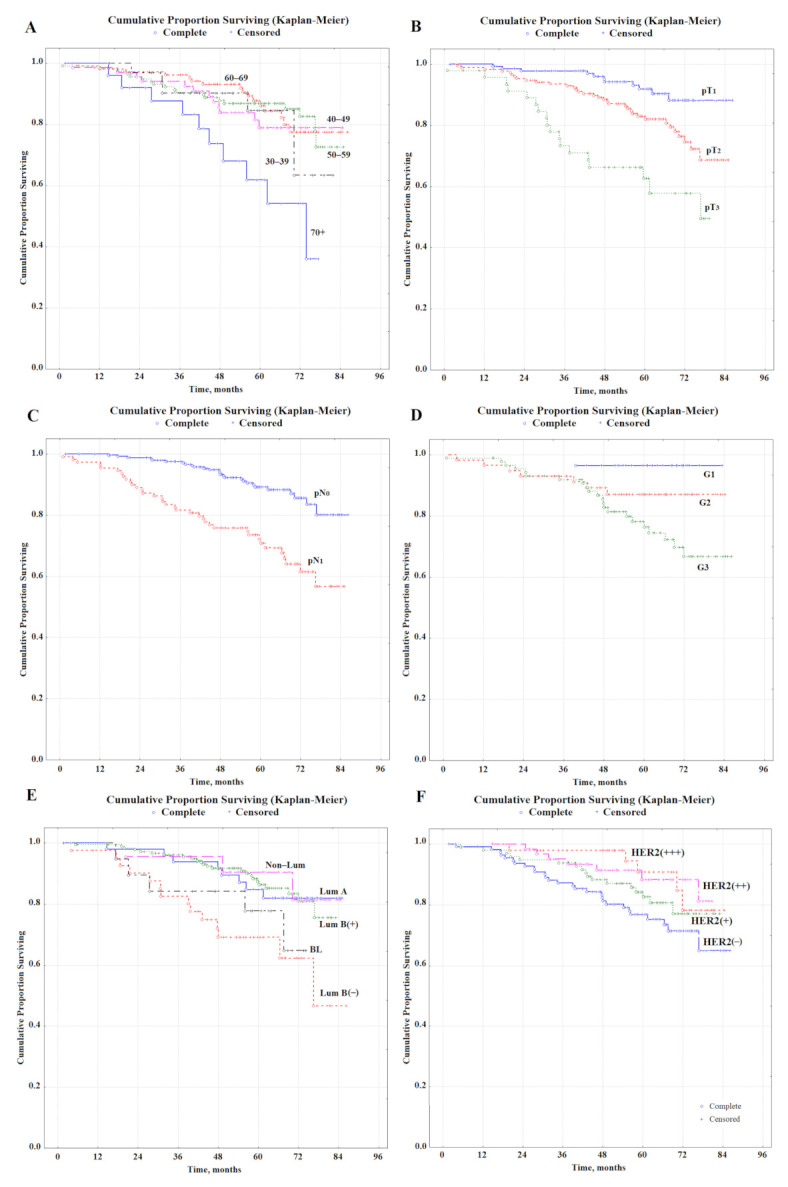
Overall survival of patients with primary resectable breast cancer depending on age (**A**), tumor size (**B**), lymph node disease status (**C**), tumor differentiation (**D**), molecular biological subtype (**E**), and HER2 status (**F**).

**Figure 2 metabolites-12-00552-f002:**
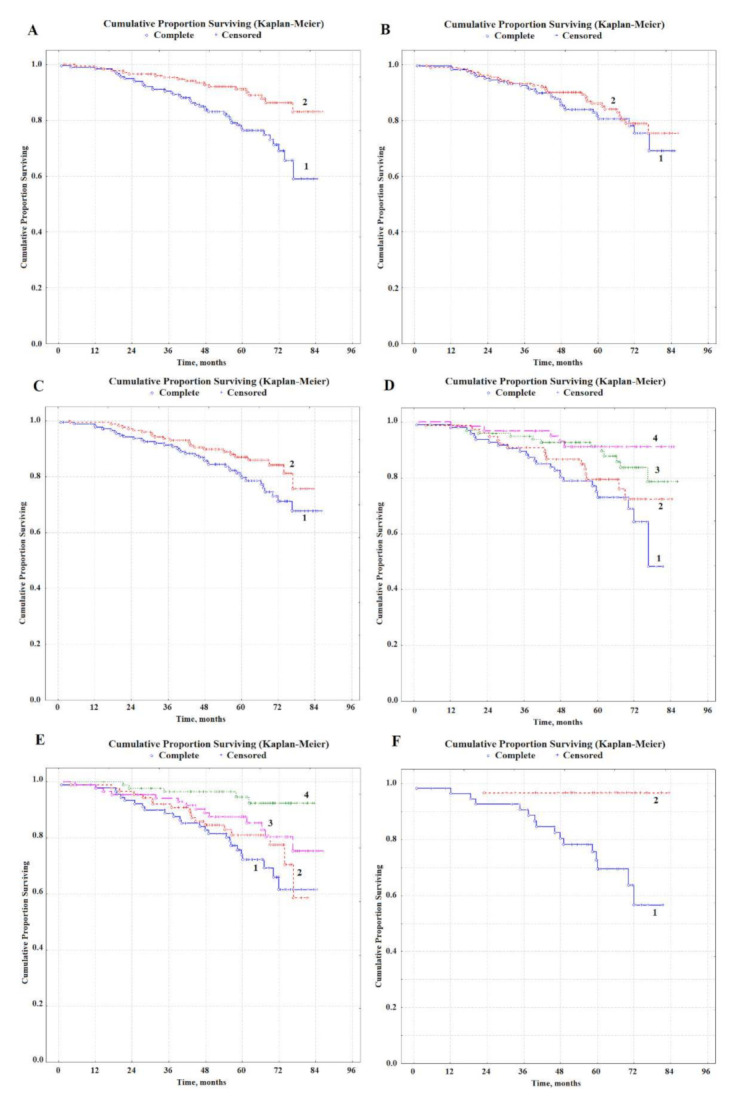
Overall survival of patients with primary resectable breast cancer depending on the biochemical composition of saliva before treatment: (**A**) unfavorable prognosis (curve 1, ALP < 71.7 U/L), favorable prognosis (curve 2, ALP > 71.7 U/L); (**B**) unfavorable prognosis (curve 1, AST > 6.33 U/L), favorable prognosis (curve 2, AST < 6.33 U/L); (**C**) unfavorable prognosis (curve 1, DC > 3.93 c.u.), favorable prognosis (curve 2, DC < 3.93 c.u.); (**D**) unfavorable prognosis (curve 1, ALP < 71.7 and AST < 6.33 U/L), favorable prognosis (curve 4, ALP > 71.7 and AST > 6.33 U/L), intermediate options (curve 2, ALP > 71.7 and AST < 6.33 U/L, curve 3—ALP < 71.7 and AST > 6.33 U/L); (**E**) unfavorable prognosis (curve 1, ALP < 71.7 U/L and DC < 3.93 c.u.), favorable prognosis (curve 4, ALP > 71.7 U/L and DC > 3.93 c.u.), intermediate options (curve 2, ALP > 71.7 U/L and DC < 3.93 c.u., curve 3, ALP < 71.7 U/L and DC > 3.93 c.u.); (**F**) unfavorable prognosis (curve 1, ALP < 71.7 U/L, AST < 6.33 U/L and DC < 3.93 c.u.), favorable prognosis (curve 2, ALP > 71.7 U/L, AST > 6.33 U/L and DC > 3.93 c.u.).

**Table 1 metabolites-12-00552-t001:** Overall survival rates depending on the clinicopathological characteristics of patients.

Category		OS, Months	HR (95% CI)	*p*-Value
Age, years	30–39, *n* = 34	56.7	1	0.06851
	40–49, *n* = 68	59.3	1.24 (0.40–3.83)	
	50–59, *n* = 117	65.4	1.12 (0.39–3.25)	
	60–69, *n* = 105	61.0	1.04 (0.35–3.07)	
	70+, *n* = 27	48.1	3.41 (1.00–11.47) *	
Menopause	No, *n* = 106	57.2	1	0.95402
	Yes, *n* = 248	61.8	1.32 (0.71–2.43)	
pT	1, *n* = 133	62.3	1	0.00000
	2, *n* = 172	60.9	2.73 (1.32–5.56) *	
	3, *n* = 47	56.3	6.88 (2.91–15.84) *	
pN	0, *n* = 245	60.7	1	0.00000
	1, *n* = 110	60.5	3.93 (2.22–6.82) *	
Grade	1, *n* = 28	64.3	1	0.06269
	2, *n* = 58	59.5	3.71 (0.44–30.98)	
	3, *n* = 88	60.8	9.00 (1.15–68.17) *	
Histological type	Ductal, *n* = 171	65.2	1	0.61737
	Lobular, *n* = 58	59.6	1.18 (0.56–2.47)	
Subtype	Luminal A-like, *n* = 50	68.0	1	0.00652
	Luminal B-like (HER2−), *n* = 41	55.1	2.72 (1.01–7.26) *	
	Luminal B-like (HER2+), *n* = 181	60.5	0.80 (0.34–1.91)	
	Non-Luminal (HER2+), *n* = 22	68.6	0.83 (0.20–3.45)	
	Basal-like, *n* = 20	58.8	1.75 (0.50–6.11)	
HER2-status	(−), *n* = 112	60.1	1	0.03198
	(+), *n* = 98	61.5	0.63 (0.32–1.24)	
	(++), *n* = 62	60.0	0.38 (0.16–0.94) *	
	(+++), *n* = 48	60.1	0.35 (0.13–0.97) *	
ER-status	(−), *n* = 49	61.7	1	0.09137
	(+), *n* = 41	60.4	1.27 (0.49–3.29)	
	(++), *n* = 57	64.9	0.83 (0.33–2.10)	
	(+++), *n* = 173	60.3	0.56 (0.25–1.23)	
PR-status	(−), *n* = 85	60.4	1	0.79103
	(+), *n* = 44	60.1	0.89 (0.35–2.25)	
	(++), *n* = 60	62.6	0.71 (0.29–1.71)	
	(+++), *n* = 131	60.2	0.85 (0.43–1.70)	

Note: *—Differences are statistically significant, *p* < 0.05; G1—highly, G2—moderately, and G3—poorly differentiated breast cancer; OS—overall survival; HR—hazard ratio; CI—confidence interval.

**Table 2 metabolites-12-00552-t002:** Overall survival rates depending on type of treatment.

Category		OS, Months	HR (95% CI)	*p*-Value
Operation status	BCS, *n* = 61	59.0	1	0.37724
	TM, *n* = 286	61.3	1.50 (0.64–3.47)	
Radiation therapy	Done, *n* = 191	60.5	1	0.00452
	Not done, *n* = 163	60.9	0.34 (0.19–0.64) *	
Chemotherapy	Done, *n* = 181	59.5	1	0.00005
	Not done, *n* = 173	61.8	0.33 (0.18–0.60) *	
Endocrine therapy	Done, *n* = 241	61.3	1	0.00413
	Not done, *n* = 113	58.3	2.18 (1.24–3.77) *	

Note: BCS—breast-conserving surgery, TM—total mastectomy; *—differences are statistically significant, *p* < 0.05; OS—overall survival; HR—hazard ratio; CI—confidence interval.

**Table 3 metabolites-12-00552-t003:** Biochemical composition of the saliva of patients with breast cancer depends on the stage.

Category		OS, Months	HR (95% CI)	*p*-Value
ALP, U/L	>71.7, *n* = 175	61.4	1	0.00243
	<71.7, *n* = 179	58.5	2.60 (1.44–4.62) *	
AST, U/L	>6.33, *n* = 174	61.5	1	0.36144
	<6.33, *n* = 163	58.5	1.13 (0.64–1.97)	
DC, c.u.	>3.93, *n* = 176	58.7	1	0.08518
	<3.93, *n* = 178	60.2	1.78 (1.02–3.08) *	
ALP + AST	>71.7, >6.33, *n* = 64	59.6	1	0.02068
	>71.7, <6.33, *n* = 98	63.3	1.80 (0.61–5.27)	
	<71.7, >6.33, *n* = 76	60.5	3.15 (1.08–9.01) *	
	<71.7, <6.33, *n* = 97	56.7	4.10 (1.47–11.18) *	
ALP + DC	>71.7, >3.93, *n* = 87	61.7	1	0.00580
	>71.7, <3.93, *n* = 87	61.0	3.15 (1.08–9.02) *	
	<71.7, >3.93, *n* = 88	56.5	4.22 (1.49–11.74) *	
	<71.7, <3.93, *n* = 91	58.9	6.21 (2.24–16.79) *	
ALP + AST + DC	Favorable, *n* = 55	62.8	1	0.01591
	Unfavorable, *n* = 29	58.9	11.49 (1.43–88.99) *	

Note: ALP—alkali phosphatase, AST—aspartate aminotransferase, DC—diene conjugates, *—differences are statistically significant, *p* < 0.05; OS—overall survival; HR—hazard ratio; CI—confidence interval.

**Table 4 metabolites-12-00552-t004:** Results of multivariate survival analysis using the Cox regression model (χ^2^ = 69.67, *p* ˂ 0.00001).

Prognostic Factors	β	Standard Error	*t*-Value	*p*-Value
Age group	0.2421	0.1363	1.7761	0.0757
pT	0.8087	0.2213	3.6550	0.0003
pN	0.9110	0.2749	3.3138	0.0009
Grade	0.7704	0.3017	2.5538	0.0107
Molecular biological subtype	0.1252	0.1184	1.0577	0.2902
HER2-status	−1.1583	0.3460	−3.3477	0.0008
ALP, U/L	−1.0105	0.3821	−2.6446	0.0082
AST, U/L	0.1803	0.3829	0.4709	0.6377
DC, c.u.	−0.4967	0.3822	−1.2997	0.1937

Note: ALP—alkali phosphatase, AST—aspartate aminotransferase, DC—diene conjugates.

**Table 5 metabolites-12-00552-t005:** The relative risk of relapse in patients with primary operable breast cancer.

Category		Relapse, *n* = 59	No Relapse, *n* = 292	HR (95% CI)	*p*-Value
Clinicopathological characteristics of patients					
Age group	30–39	9	28	1	0.69523
	40–49	16	51	0.98 (0.38–2.48)	
	50–59	19	98	0.60 (0.25–1.48)	
	60–69	13	91	0.44 (0.17–1.15)	
	70+	2	25	0.25 (0.05–1.26)	
Menopause	No	40	84	1	0.00526
	Yes	19	208	0.19 (0.11–0.35) *	
pT	1	10	123	1	0.00032
	2	30	141	2.62 (1.23–5.51) *	
	3	19	28	8.35 (3.47–19.50) *	
pN	0	26	216	1	0.00079
	1	33	76	3.61 (2.02–6.34) *	
Grade	1	4	24	1	0.46751
	2	8	50	0.96 (0.27–3.48)	
	3	16	71	1.35 (0.41–4.40)	
Histological type	Ductal	32	137	1	0.98523
	Lobular	14	44	1.36 (0.67–2.76)	
Molecular biological subtype	Luminal A-like	11	39	1	0.72153
	Luminal B-like (HER2−)	8	33	0.86 (0.31–2.37)	
	Luminal B-like (HER2+)	30	149	0.71 (0.33–1.55)	
	Non-Luminal (HER2+)	2	20	0.35 (0.07–1.75)	
	Basal-like	4	15	0.95 (0.26–3.41)	
HER2-status	(−)	24	87	1	0.12697
	(+)	16	81	0.72 (0.36–1.44)	
	(++)	10	52	0.70 (0.31–1.57)	
	(+++)	7	39	0.65 (0.26–1.63)	
ER-status	(−)	8	39	1	0.56214
	(+)	11	30	1.79 (0.64–4.94)	
	(++)	9	48	0.91 (0.32–2.57)	
	(+++)	29	142	1.00 (0.42–2.34)	
PR-status	(−)	16	69	1	0.64852
	(+)	6	36	0.72 (0.26–1.99)	
	(++)	10	50	0.86 (0.36–2.05)	
	(+++)	25	105	1.03 (0.51–2.05)	
**Type of treatment**					
Operation status	BCS	5	56	1	0.16325
	TM	51	235	2.43 (0.93–6.29)	
Radiation therapy	Done	41	150	1	0.00289
	Not done	18	145	0.45 (0.25–0.83) *	
Chemotherapy	Done	42	139	1	0.00117
	Not done	17	156	0.36 (0.20–0.66) *	
Endocrine therapy	Done	38	203	1	0.62547
	Not done	21	92	1.22 (0.68–2.18)	
**Biochemical indicators of saliva**					
ALP, U/L	>71.7	22	163	1	0.00524
	<71.7	36	129	2.07 (1.16–3.66) *	
AST, U/L	>6.33	29	142	1	0.69441
	<6.33	28	133	1.03 (0.58–1.82)	
DC, c.u.	>3.93	26	148	1	0.45597
	<3.93	33	144	1.30 (0.74–2.28)	
ALP + AST	>71.7, >6.33	16	86	1	0.14965
	<71.7, <6.33	23	67	1.85 (0.90–3.73)	
ALP + DC	>71,7, >3.93	10	81	1	0.00124
	<71.7, <3.93	21	61	2.79 (1.22–6.27) *	
ALP + AST + DC	Favorable	7	45	1	0.08963
	Unfavorable	14	37	2.43 (0.89–6.57)	

Note: *—The differences are statistically significant, *p* < 0.05. ALP—alkali phosphatase, AST—aspartate aminotransferase, DC—diene conjugates.

## Data Availability

The data presented in this study are available on request from the corresponding author. The data are not publicly available because they are required for the preparation of a Ph.D. thesis.

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
