# Peer review of "Prognostic Value of Salivary Biochemical Indicators in Primary Resectable Breast Cancer"

_metabolites, 2022, doi:10.3390/metabo12060552_

Round 1
Reviewer 1 Report
In this study, the authors collected 355 saliva samples from primary resectable breast cancer patients to examine the composition of saliva before treatment on overall survival and the risk of recurrence. They found that ALP, AST and DC could act as prognostic markers. And the Significant risk factors for recurrence were also related with salivary ALP activity and DC levels. The conclusions were supported by some data. However, the results presented do not convincingly support their conclusions and several issues need to be addressed.
1. The introduction part is not closely related with this paper, and should also have some information on ALP, AST and DC, please reorganize this part.
2. According to Table 3, both ALP and DC combined with AST would attenuate the significant indications in the patients, why the authors still think AST is a good prognostic marker?
3. ALP, AST and DC were only in saliva or also could be detected in blood samples? If so, did they show the similar results?
4. It seemed that the results did not fully support the conclusions that ALP, AST and DC were prognostic markers for primarily resectable breast cancer, due to the levels of AST did not show significant differences.
5. From lines 210-213, the authors think that the activities of a number of enzymes in blood serum and saliva do not always coincide, so if this happened in a patient, which results should be reliable?
6. Do the authors think that these prognostic markers can be used in other types of cancer, such as lung cancer?
7. Please provide the biochemical analysis of saliva samples.
Author Response
The authors express their deep gratitude to the reviewers for their attentive attitude to the manuscript and valuable comments. Responses to the comments of the reviewers are in the attached file.

Reviewer 2 Report
The authors of the article entitled “Prognostic value of salivary biochemical indicators in primary 2 resectable breast cancer” present that three biochemical indicators of saliva (ALP, AST, DC) can act as prognostic markers in primary 2 resectable breast cancer. I see several major problems with the current submission.
1. Introduction: Too much information about the unexamined in this manuscript prognostic factors. In turn, there are no literature data about identified by authors three biochemical indicators which can act as prognostic markers, alkaline phosphatase (ALP), aspartate aminotransferase (AST) and diene conjugates (DC). There are lack of any information about diene conjugates (DC) in all text. I recommend rebuild this section and complete discussion.
2. Results: There are conflicting opinions in the text. In the abstract section we can find “Favorable prognostic factors were ALP activity above 71.7 U/L, AST activity above 6.33 U/L and DC level above 3.93 c.u” Further the authors write "AST activity had no independent prognostic value (Fig. 2b); however, in combination with ALP activity, it made it possible to refine the prognosis of breast cancer (Fig. 2e)" It is not true that ALT may act as a prognostic factor. The value of p for this marker is not statistically significant (p=0.36144). See Table 3, Figure 2. Please explain.
The sentence “Basally similar breast cancer had an unfavorable prognosis; however, due to the small number of this subgroup, the differences with other groups were not statistically significant” is not clear. Please modify.
3. Discussion: This section is poor and does not relate to the literature on the subject (e.g. doi: 10.1016/j.clgc.2016.08.023, doi: 10.1007/s13205-012-0113-1). Please complete.
4. The graphs in Figures 1 and 2 should be better described. It is not known what parameter they describe. This makes it difficult to analyze the results.
5. Did the authors analyze the levels of selected markers also in the plasma/serum? It would be worth comparing their values with those obtained in saliva.
In my opinion manuscript in current version is not suitable for publication in Metabolites. I suggest the authors to revise the manuscript according to the above mentioned.
Author Response

(The authors gave the same response as above.)

Round 2
Reviewer 1 Report
In this study, the authors collected 355 saliva samples from primary resectable breast cancer patients to examine the composition of saliva before treatment on overall survival and the risk of recurrence. They found that ALP, AST and DC could act as prognostic markers. And the Significant risk factors for recurrence were also related with salivary ALP activity and DC levels. The conclusions were supported by solid data. And can provide some useful messages in the prognosis of primary resectable breast cancer.
Reviewer 2 Report
The authors have modified the article in line with the comments. In my opinion the current version is sufficient for publication in Metabolites.